# Activity of Apo-Lactoferrin on Pathogenic Protozoa

**DOI:** 10.3390/pharmaceutics14081702

**Published:** 2022-08-15

**Authors:** Magda Reyes-López, Gerardo Ramírez-Rico, Jesús Serrano-Luna, Mireya de la Garza

**Affiliations:** Departamento de Biología Celular, Centro de Investigación y de Estudios Avanzados del Instituto Politécnico Nacional (CINVESTAV-IPN), Mexico City 07360, Mexico

**Keywords:** lactoferrin, apo-lactoferrin, parasitic protozoa, pathogens

## Abstract

Parasites and other eventually pathogenic organisms require the ability to adapt to different environmental conditions inside the host to assure survival. Some host proteins have evolved as defense constituents, such as lactoferrin (Lf), which is part of the innate immune system. Lf in its iron-free form (apo-Lf) and its peptides obtained by cleavage with pepsin are microbicides. Parasites confront Lf in mucosae and blood. In this work, the activity of Lf against pathogenic and opportunistic parasites such as *Cryptosporidium* spp., *Eimeria* spp., *Entamoeba histolytica*, *Giardia duodenalis*, *Leishmania* spp., *Trypanosoma* spp., *Plasmodium* spp., *Babesia* spp., *Toxoplasma gondii*, *Trichomonas* spp., and the free-living but opportunistic pathogens *Naegleria fowleri* and *Acanthamoeba castellani* were reviewed. The major effects of Lf could be the inhibition produced by sequestering the iron needed for their survival and the production of oxygen-free radicals to more complicated mechanisms, such as the activation of macrophages to phagocytes with the posterior death of those parasites. Due to the great interest in Lf in the fight against pathogens, it is necessary to understand the exact mechanisms used by this protein to affect their virulence factors and to kill them.

## 1. Introduction

### 1.1. Parasitism as a Way of Life

Parasitism is the process by which a host-dependent relationship is established by an organism to gain access to nutrients and survive at the expense of a host. Parasitism emerged early in evolution, and parasites have adapted exquisite mechanisms for living at the cost of the host. Parasitism is characterized by a unilateral relationship in which one of the members (the host) contributes and the other member (the parasite) receives benefits without giving something in return. A non-pathogenic parasite successfully receives these benefits without significantly damaging or killing the host.

Parasitism carried out by pathogenic organisms is a detrimental relationship in which the parasite substantially damages or kills the host [1,2,3,4,5,6]. A disadvantage of the parasitic lifestyle is the necessity of a repertoire of adaptations, including the availability and use of iron proteins from the host to assure entry–exit from the host and to counter immune responses to prevent clearance [7]. The ability of parasites to manipulate host responses and make use of different organs, tissues, cells, and fluids of the body and establish infection is frankly sinister; however, despite being an important problem for human health, the molecular mechanisms that are involved in the host–parasite relationship are not fully understood [8].

### 1.2. Opportunistic Free-Living Amoebae as Pathogenic Protozoa

Free-living amoebae are not parasites, but they can be pathogenic to a host that they eventually infect. Several species have been described that can cause severe damage to humans and animals.

## 2. Activities of Lactoferrin on Pathogenic Protozoa

The interactions of lactoferrin (Lf) with pathogenic protozoa comprise two types, depending on the iron content of the Lf. Apo-Lf, an iron-free molecule, can be a parasiticide, and holo-Lf, a molecule laden with one or two ferric-iron ions, can be an iron source for several species of parasites, although it can also be a parasiticidal protein for a few parasites.

### 2.1. Apo-Lf Can Damage and Kill Pathogenic Protozoa

Lactoferrin is a non-heme glycoprotein from the innate immune system of mammals. Lf is strategically situated in the different mucosae but is especially abundant in colostrum and milk. In these fluids, Lf contains a very low concentration of iron and is called apo-Lf. In addition, Lf is produced by neutrophils and secreted in the secondary granules of these immune cells at the sites of infection to chelate iron and prevent pathogen access to this essential element, often killing them. This review particularly focuses on the role of apo-Lf and its peptide derivatives in the inhibition of pathogen growth, its high affinity for biological membranes, and lethal effect by disruption of essential membrane functions.

Parasites and free-living protozoa interact with endogenous Lf in mucosae and blood, in which Lf can be a microbiostatic and/or microbiocidal component. In experimental studies, when apo-Lf is added to parasite culture media or given to animal models to assay its biological activity, it has been shown to have a biocidal effect on some parasite species.

In 1992, Bellamy and his coworkers [9] observed that substantial amounts of Lf enter the gastrointestinal tract of mammals as a component of saliva, colostrum, and milk, and ingested Lf appears to have a significant role in the protection of neonates from infectious diseases. Posteriorly, they have shown that the active peptides generated by pepsin cleavage of human and bovine Lf present bactericidal properties more potent than undigested Lf. Moreover, this finding has led to the identification of the bactericidal domain of Lf near the N-terminus of the Lf molecule. These peptides, called Lactoferricins (Lfcins), are well known in their bovine and human forms [10,11]. Lfcins can pass through the microbial cell membrane and nuclear envelope, suggesting that nucleic acids are a potential target for Lfcins. Lfcins are strongly hydrophobic with positively-charged surfaces [12]. Lfcin from bovine is LfcinB17–41, which forms a looped structure through an intramolecular disulfide bond that forms the cyclic structure, which is important for its greater antibacterial activity [10,11]; human Lfcin [9] is LfcinH1-47, which is formed by two subfragments connected by disulfide bonds between cysteine residues 1–11 and the cyclic residues 12–47 [10].

On the other hand, based on such common features of antimicrobial peptides, a new peptide was synthesized, the lactoferrampin (Lfampin268–284), from bovine Lf with a broad spectrum of antibacterial activity, which could be isolated using enzymatic activity from bovine Lf. Additionally, human Lfampin was also synthesized (269–285) [10,11]. A fusion peptide between Lfcin 17–30 and Lfampin 265–284 was produced and named Lactoferrin-chimera (Lfchimera)—it was formed by 35 amino acids. The binding of these peptides into one molecule resulted in greater antimicrobial activity than each of its peptides and showed a great synergistic activity with different antibiotics [10].

Lfcins, Lfampin, and Lfchimera have been tested against different organisms and they have shown anti-bacterial, anti-parasitic, anti-fungal, anti-viral, and anti-inflammatory activities, and have more more potent antimicrobial actions than the intact protein [10,11,12].

Thus, assays have been performed using diverse Lfcins and Lfampin. These peptides have been probed against bacterial pathogens, and they are more efficient than Lf in destroying them [9,13]. Since Lf-derived peptides are produced by gastric pepsin cleavage of Lf, they can be found in vivo [13,14].

The mechanisms by which apo-Lf inhibits parasite growth and affects parasite virulence are the theme of a vast literature that describes it’s in vitro or in animal models efficacy, which is reported below. It has been demonstrated that apo-Lf interacts with protozoan membrane cell constituents, such as phospholipids and proteins, destabilizing the membrane and leading to parasite death, as we describe below. The parasiticidal effects of Lf from human, bovine, and buffalo origins have primarily been studied.

### 2.2. Holo-Lf Can Be an Iron Source for Pathogenic Protozoa

Parasitic and free-living protozoa can be pathogenic to animals and humans, and they can develop a series of mechanisms to survive inside the host. Iron is a toxic element but is vital for all forms of life; thus, it is particularly important for both hosts and parasites. Due to its toxicity, iron is captured in proteins, restricting its availability to all pathogens. In fact, the free-iron concentration in fluids is 10^−18^ M, a scarce quantity for the parasite’s requirements; proteins such as Lf are devoted to iron withholding, contributing to prevention of the growth of microorganisms. One of the virulence mechanisms of invading parasites is to acquire iron from host iron-containing proteins, including holo-Lf. Apo-Lf in mucosae can acquire iron from the diet and in blood it can be acquired from ferric transferrin, and the resulting holo-Lf can be an iron source for some pathogens. To use the iron from holo-Lf, parasites have developed specific receptors and secreted proteases that cleave and degrade holo-Lf. In this review, we focus on the effect of exogenous apo-Lf to combat pathogenic protozoa. The use of holo-Lf as an iron source by these microorganisms has been reviewed elsewhere [15].

## 3. Effect of Lactoferrin on Pathogenic Parasites

Due to the emergence of pathogenic protozoa resistant to the drugs used as treatments and the side effects that they cause to the patients, it is necessary to develop other products to combat them. Normally, the treatment for pathogenic protozoa infection is long and has several undesirable side effects, this causes the treatment to be abandoned with the resultant appearance of drug refractory pathogens. In *Giardia duodenalis*, for example, the recent emergence of strains resistant to the treatment rapidly increased in just a few years [16,17]. Metronidazole is a common first-line treatment for giardiasis [16,18], amoebiasis [19], and trichomoniasis [20]. Often, the conventional drugs present low efficacy and poor safety. There are no vaccines against major parasitic infections and drugs are the only treatment option.

Natural remedies usually have the advantage of being innocuous, and parasites could show sensitivity without further resistance. These include extracts, fractions, pure compounds, or minerals that are biosynthesized in nature. There are primary and secondary metabolites. Primary metabolites are conserved compounds and necessary for life, while secondary metabolites are not essential for growth but indispensable for survival. Commonly, secondary metabolites participate in defense, protection, and signaling [21]. Among several products used, the natural product, Artemisin, an antimalarial drug isolated from *Artemisia annua*—a plant used in traditional medicine [22]—has been used in the treatment of patients with multidrug-resistant *Plasmodium falciparum* parasite [23].

Lf is a natural product that has been studied extensively over the past few decades. It is a multifunctional protein best known for its ability to bind iron, which eventually led to the discovery of its antimicrobial activity—and is considered an important host defense molecule. There are few studies of Lf against pathogenic protozoa, and they were mainly performed in vitro and in animal models; in the following, we describe some examples of the assays conducted (Figure 1).

### 3.1. Cryptosporidium spp.

*Cryptosporidium* is part of the Coccidia group of protozoa, which are characterized by the ability to reproduce asexually and sexually. However, unlike other coccidian parasites, *Cryptosporidium* organisms infect only the microvilli of the intestinal epithelium—not other histological layers of the mucosa. Human infectious processes have been reported in association with more than 15 species of *Cryptosporidium.* The two main species of Cryptosporidium that cause pathology are *C. hominis* and *C. parvum*. *C. parvum* is often identified in rural areas, as it is associated with cattle and other animals [24,25]. Exposure to animals, particularly cats and cattle, is linked with an increased risk of infection, and children and the elderly are diagnosed with cryptosporidiosis the most. Patients with acquired (human immunodeficiency virus, chemotherapy, immunosuppressive drugs) or congenital (hypogammaglobulinemia, IgA deficiency) immunodeficiency are at great risk of infection and prolonged or severe illness.

*Cryptosporidium* can persist in the environment as an oocyst containing four sporozoites, which are the infective form of the parasite. After exposure, the incubation period varies from two to ten days. Once ingested, the oocysts travel to the small intestine, and excystation releases the sporozoites. These parasite cells settle inside the walls of the small intestine and multiply asexually within extracytoplasmic parasitophorous vacuoles. Each cell reproduces in large numbers, producing oocysts that can be passed in the feces and persist in the environment [26].

In 2012, the effects of various forms of Lf on *C. parvum* were studied, including Lf, hydrolyzed Lf [27], and LfcinB4-14. The three Lfs were of bovine origin and were administered at a dose of 10 µg/mL. Each sample was incubated with 2 × 10^5^ purified sporozoites. Hydrolyzed Lf and LfcinB showed a parasiticidal effect that affected the viability and decreased the infectivity of the pathogen. However, Lf did not show statistically significant results compared to the control [28]. In contrast, in 2017, the effect of Lf on the oocyst phase was tested, and no effect was found toward this resistant phase of *Cryptosporidum*. In addition, Lf did not show effectiveness in the intracellular stage, so the effect of Lf is limited to extracellular sporozoites [29]. These findings are interesting, and the use of Lf can be suggested as a preventive therapy for this disease, mainly for immunocompromised patients and children from low-income areas.

### 3.2. Eimeria spp.

The phylum Apicomplexa involves many coccidia that cause coccidiosis and comprise the most important infectious process in poultry production. *Eimeria* is the largest genus in this phylum, with over 1800 species described [30]. *Eimeria* shares some similarities with other coccidia genera, such as *Cystisospora*, *Cyclospora*, *Epieimeria*, *Hammondia*, *Karyolysus*, *Neospora*, *Sarcocystis*, and *Toxoplasma*, but are less closely related to *Cryptosporidium* [31]. *Eimeria* are obligate intracellular parasites in all classes of vertebrates with absolute host and tissue specificity; the disease can be easily transferred between conspecific hosts [32].

The biological life cycle of *Eimeria* comprises several phases: gametogony (sexual) and schizogony (asexual) occur in the host, while sporogony (asexual) occurs outside the host. Susceptible hosts are infected after ingestion of sporulated oocysts, which have two to four sporocysts. From each sporocyst, two motile sporozoites are released that continue their way to the host intestinal epithelium and invade it to form nonmotile trophozoites. Intracellular sporozoites then transform into spheroidal schizonts and continue their asexual development or nuclear division to form merozoites by merogony. The merozoites released from schizonts can re-invade new epithelial cells or become macro- and microgametes, which, when fused, result in zygotes and oocysts. The number of merozoite generations varies depending on the species of coccidia [33,34].

Although there are many reports of Lf activity toward different microorganisms, there are few tests toward parasites, especially toward coccidia, for which there is only one report. In 2001, Omata et al. tested the effectiveness of bLfcin against *E. stiedai* in vivo (rabbits and mice) and in vitro (rabbit hepatobiliary cells and mouse embryonic cells) using two different concentrations of Lfcin, 100 or 1000 ug/mL, and *E. stiedai* sporozoites at an inoculum of 10^6^ parasites/ml and incubated for 18 h. Subsequently, the sporozoites were fixed to observe the effect, 10^5^ from each treatment were inoculated intravenously, and the animals were monitored for 60 days after inoculation. Sporozoites treated with 1000 µg/mL bLfcin showed less infectivity and less penetration into host cells than untreated sporozoites. The rabbits inoculated with sporozoites at this dose presented a lower number of oocysts, the livers had fewer abscesses and they did not present inflammation of the bile ducts [35]. As the authors noted, more experiments are needed to determine a more effective dose and test different routes of administration and other forms of Lf.

### 3.3. Entamoeba histolytica

*Entamoeba histolytica* is a common human intestinal parasitic protozoan that produces amoebiasis, the third largest cause of death by parasites. It is an important human infection transmitted by the ingestion of contaminated food or water. It results in approximately 100,000 deaths and 50 million people infected worldwide per year [36]. As demonstrated by its name, this parasite can produce massive tissue destruction; trophozoites (amoebas) invade the intestinal mucosa, causing dysentery, fever, and abdominal pain. In some cases, amoebas can spread to the liver, lungs, and brain and can cause death if not treated. Infection of the large intestine and liver are the main forms of amoebiasis, resulting in intestinal amoebiasis (IA) or liver amoebiasis, also called amoebic liver abscess (ALA), respectively. However, no more than 10–20% of *E. histolytica* infections result in disease, which includes self-limiting colitis, invasive colitis, and extraintestinal infection [37,38,39]. Adherence, contact-dependent cytotoxicity, phagocytosis, trogocytosis (a special process in which amoebas ingest pieces of intact living cells), and proteases (among other secreted products) form the arsenal of virulence mechanisms and factors that contribute to the damage [39]. In addition, there are other *Entamoeba* species, such as *E. dispar*, which is a low-pathogenicity species that does not cause disease, but it is morphologically identical to *E. histolytica* when cysts are observed with a microscope; thus, an appropriate diagnosis is essential.

More than 40 years ago, Murray [40] observed that milk-drinking African people did not show infections with *E. histolytica*. In Maasai people with a diet based on milk, lactobacilli are predominant in the colon, and these bacteria contain very little iron concentration. Therefore, even if amoebas ingest lactobacilli, the iron concentration is not enough to support the growth of this parasite. The intestinal iron concentration is responsible for controlling the growth of the parasite, independent of the iron concentration in the body. In contrast, other tribes with a high-iron diet presented recurrent amoebic infections [40]. This beneficial property of milk was observed first in animals infected with malaria parasites, which were fed bovine milk, and the infection diminished [41]. Most iron in milk is bound to Lf or transferrin (Tf), and both proteins were saturated up to 30% (human milk Lf is generally saturated about 3–5%) [42], taking away free iron and avoiding amoebae or other pathogens that can bind iron. The very low quantity of iron in milk and the presence of Lf prevent parasites from fulfilling their iron necessity, making it inaccessible to invade the host. Trophozoites of *E. histolytica* have high requirements for iron in culture (50–70 µM) [43,44].

Partly and totally iron-saturated Lf are more resistant to digestion than apo-Lf, which can maintain the ability to bind iron in the gut. In the large intestine, amoebas may find other iron sources, such as colonic desquamation cells or red blood cells (RBCs), when they destroy the tissues. In experimental studies, it was observed that amoebas in a low-iron medium do not present invasive qualities (virulence factors), and the low iron could be insufficient to satisfy their iron necessity. Thus, the amoebicidal effect of human milk has been studied in cultures, obtaining approximately 90% lethality [45]. That work discarded the participation of the secretory immunoglobulin A (sIgA) present in milk, among other antibacterial proteins, despite the high concentrations found in the milk.

To highlight the mechanisms by which milk Lf is an amoebicide, our group explored the effect of different concentrations of apo-Lf on parasites in axenic cultures and animal models of amoebic infection. bLf and hLf were amoebicidal at 100 µM after 1 h of incubation (10% viability). Interestingly, apo-Lf was observed on the amoeba surface after 5 min of interaction, after which the trophozoite membrane was permeabilized and amoebas died. Ferric iron and some divalent cations reversed the Lf killing effect, which was modulated by culture age, pH, and temperature [46]. It is known that human milk protects breast-fed children from bacterial pathogens and the main proteins that can provide this protection are Lf, sIgA, and lysozyme [47,48,49]. In addition, synergistic activity between Lf and lysozyme has been reported in *Staphylococcus epidermidis* [47], and between lactoperoxidase and Lf in *Acinetobacter baumannii* [50]. We separately assayed the effect of human or bovine milk Lf and sIgA or egg chicken lysozyme, which is very similar to the human lysozyme, and the three proteins showed amoebicidal activity. However, apo-Lf showed major amoebicidal action, and the amoebicidal activity of hLf and bLf increased when amoebas were incubated in the absence of iron. Moreover, IgA, apo-Lf, and lysozyme (100 µM each) were simultaneously incubated with amoebas for 1 h, and then we observed their morphology. These proteins caused rearrangements and disruption in the lipid pattern after being bound to the amoeba membrane. Although the susceptibility of *E. histolytica* to the three proteins was assayed in vitro, the parasites might also be affected in vivo, since these proteins can be found in the large intestine where *E. histolytica* infection occurs [51]. We also tested whether the Lf-derived peptide, Lfcin4-14, is amoebicidal by incubating amoebas with different concentrations of the peptide; this peptide was observed in trophozoites and reached the nucleus, destroying the parasites [46]. Bolscher et al. produced a fusion product of Lfcin17-30 and Lfampin265-284, named Lfchimera [52]; when tested, all could cause the death of amoebas in axenic cultures, although Lfchimera exhibited a major amoebicidal effect [53].

Synergy between bLf and antifungals against *Candida albicans* was reported early on [54]. We explored whether a synergistic effect would be observed in axenic cultures between Lf or LfcinB and metronidazole, the drug of choice in the treatment against *E. histolytica*. The reasoning was that metronidazole is a highly toxic drug but is currently an effective treatment against amoebiasis; if combined with Lf, the metronidazole toxic dose for ALA could generally be diminished [51]. In this work, a co-active amoebicidal effect was observed, even when the metronidazole concentration was diminished. Thus, a low metronidazole dose is recommended since it could reduce the toxicity and adverse side effects of treatments for patients. It is important to note that the Lf killing effect was observed in the absence of iron, since when the iron was added, the Lf amoebicidal property was prevented. Lf and its derivatives were shown to be amoebicides depending on their concentration, and the effect was modulated by the temperature, pH, and age of culture. This study concludes that Lf or bLfcin, in combination with metronidazole at low doses, can be used as a therapeutic option due to the diminished side effects and the risk of producing amoebic drug resistance [51]. Metronidazole-resistant variants of *E. histolytica* have not been reported in the clinic but have been obtained in amoeba cultures [19].

The therapeutic outcome of bLf was tested by our group in an intestinal amoebiasis model in C3H/HeJ mice (a strain of mice highly susceptible to *E. histolytica*) [55] with daily oral administration of 20 mg/kg bLf. After only one week of treatment, Lf eliminated the amoebic infection by activating the production of anti-amoeba IgA antibodies that could block the adherence of amoebas to the gut epithelium. In addition, Lf competes for iron and shows amoebicidal activity by disrupting the parasite membrane [56,57]. To determine the effect of Lf against hepatic amoebiasis, apo-bLf was probed in a Syrian hamster model of ALA. Hamsters were intragastrically treated with metronidazole or Lf or with a mixture of low quantities of each [58]. Hamsters treated with Lf combined with metronidazole showed no clinical signs of disease, even with a third of the typical metronidazole concentration used to cure ALA in these animals. We found only a very small lesion percentage of 0.63% in the liver compared with untreated animals, which presented a liver injury of 63%. Practically all animals treated with the combination of Lf and metronidazole were healed in 8 days, and liver function and blood cells approached normal levels in hamsters receiving Lf treatment. In this work, the beneficial use of Lf was demonstrated for potential therapeutic use alone or with metronidazole to treat amoebiasis.

Finally, as an initial approach to understand the amoebicidal effect of Lf, Lfcin B, Lfcin17-30, and Lfampin were used to analyze how these peptides solve an entrenched amoebic intercaecal infection in C3H/HeJ mice. All peptides were internalized by a clathrin-independent route. The internalization pathway mainly requires PI3K signaling. The actin cytoskeleton also participates for two of the peptides. Lf peptides bind to several amoebic proteins and cholesterol, leading to amoebic lysis. It was observed that Lfampin interacts with amoebic membranes through cholesterol; this effect might lead to parasite lysis by causing loss of membrane integrity [59].

The antimicrobial activity of the multifunctional protein Lf and its derived peptides demonstrated in in vitro and in vivo studies might be considered an option for the treatment of amoebic intestine and liver disease. However, it is imperative to gain a profound understanding of the mechanisms of action of Lf, a natural and innocuous protein for the host, to determine its biological properties against *E. histolytica*.

### 3.4. Giardia duodenalis

*Giardia duodenalis* (also known as *Giardia intestinalis* or *Giardia lamblia*) is a protozoan flagellate (Diplomonadida). This protozoan was initially named *Cercomonas intestinalis* by Lambl in 1859 and renamed *Giardia lamblia* by Stiles in 1915 in honor of A. Giard and F. Lambl. However, many consider the name *Giardia duodenalis* to be the correct taxonomic name for this protozoan (https://www.cdc.gov/healthywater/surveillance/giardiasis/giardiasis-2019, accessed on 6 July 2022). *G. duodenalis* is the causal agent of human giardiasis, a disease of the small intestine that is characterized by diarrhea. It is a worldwide disease that especially affects children in developing countries. The *Giardia* life cycle consists of two forms, cyst and trophozoite (giardia); the infective cysts are transmitted by contaminated food and through the fecal–oral route. The WHO has estimated approximately 100 million cases of giardiasis at any time, contributing to 2.5 million deaths annually from diarrheal disease [19]. When ingested, cysts become trophozoites that colonize the small intestine, affecting its ability to absorb fat, lactose, and vitamins A and B12, which may lead to weight loss and cause malnutrition, a characteristic of *G. duodenalis* infection. Malnutrition in early childhood is associated with poor cognitive function and stunting [60]. Recently, Ferreira et al. measured the hematological profiles in infected gerbils and demonstrated that malabsorption as well as chronic inflammation may be implicated in iron deficiency anemia in giardiasis [61]. *Giardia* is a flagellated unicellular eukaryote parasite [60,62].

Metronidazole and its derivative tinidazole are considered drugs of choice against giardiasis. However, clinical resistance prevalence levels of 20% and recurrence rates as high as 90% have been reported; in addition, resistant parasites have been isolated from patients showing refractoriness to metronidazole and furazolidone [19]. Turchany et al. initiated research on the use of Lf and its N-terminal peptides as giardicides and demonstrated the killing effect on parasites. In their experiments, both Lf and its peptides bound to the parasite surface, and interestingly, the giardicidal concentration of Lf and its peptides to kill trophozoites was physiologically found in the duodenal lumen. On a molar basis, the peptides from bovine and human origin were significantly more potent than the corresponding whole Lf molecule. Moreover, attached and lumen trophozoites might interact with Lf in the small intestine. In addition, giardia showed differences in susceptibility to death by Lf depending on the growth phase. Ferric iron and some divalent cations present in the small intestine, such as Mg and Ca, protected the parasites from the effects of Lf [63].

Regarding the effects of Lf on the structure of the trophozoite, two years later, the aforementioned team of researchers demonstrated that Lf and its peptides cause remarkable and complex morphologic changes in the giardia plasmalemma, internal membranes, and cytoskeleton, and increased the electron density of the lysosome-like peripheral vacuoles [64]. In the comprehensive work of Frontera et al., the endocytosis of bLf and bLfcin (a synthesized peptide of 25 residues) by *G. duodenalis* and their effects on cell homeostasis were analyzed. Both bLf and bLfcin were localized in peripheral vacuoles (PVs), which are organelles located below the plasma membrane that function as endosomes and lysosomes in *G. duodenalis*. This cellular process was carried out by a parasite receptor and led to the cessation of cell growth. bLf and its derivative peptide caused morphological changes in the trophozoites, which ultimately produced immature cysts [60]; encystment allows for many microorganisms to adopt a latent, highly-resistant phase and, when conditions are improved, they change to a reproductive stage. Thus, the disruption of the *G. duodenalis* cyst-stage progress could be a strategic target for the development of new approaches against the transmission of this parasite [65]. Recently, Aguilar-Díaz et al. demonstrated that the exposure of giardias to Lf synthetic peptides, Lfcin17-30, Lfampin265-284, and Lfchimera, were correlated with an increase in cytoplasm granularity and vacuolization, formation of pores, and extensive membrane disruption. At low concentrations, all the peptides had remarkable parasiticidal action in the trophozoites, and the effect was more evident when they were combined with metronidazole or albendazole. Regarding the mechanism of action of the Lf peptides, they observed programmed cell death in *G. duodenalis* trophozoites [66].

Concerning the use of Lf in patients, Ochoa et al. [67] conducted a randomized, double-blind, placebo-controlled community-based 9-month trial to compare supplementation with oral bLf (0.5 g daily) vs. placebo for the prevention of diarrhea in Peruvian children (Lf group *n* = 146) aged 12–36 months. A comparison of overall diarrhea incidence and prevalence rates revealed no significant difference between the two groups. However, there was a lower prevalence of colonization with *Giardia* spp. and better growth of children in the group treated with Lf. The mean number of samples that were positive for *Giardia* species per child was lower in the Lf group than in the placebo group, with a longer mean duration of *Giardia* carriage in the placebo group than in the Lf group (4.6 vs. 3.1 months).

The above in vitro and in vivo studies allowed for us to propose bLf or hLf and their peptides to be used in the treatment of giardiasis.

### 3.5. Leishmania spp.

Leishmaniasis is caused by more than 20 species of the genus *Leishmania*, a parasitic protozoan with a prevalence of 12 million cases and an approximate annual incidence of 0.5 million cases of visceral leishmaniasis. It is considered a major neglected tropical disease that is potentially fatal if left untreated [68]. The most common forms of the disease are cutaneous and visceral leishmaniasis. The cutaneous type causes skin sores. The visceral type affects the internal organs, such as the spleen, liver, and bone marrow. People with this disease usually have fever, weight loss, and an enlarged spleen and liver. Leishmaniasis is transmitted by the bite of an infected female sandfly, which needs a blood meal to produce eggs. More than 90 species of sandflies are known to transmit this parasite. The epidemiology of leishmaniasis depends on the characteristics of the parasite and sandfly species, the ecology of the places where it is transmitted, the previous and current exposure of the human to the parasite, and human behavior. There are approximately 70 animal species, including humans, that are natural reservoirs of *Leishmania*. Visceral, cutaneous, and mucocutaneous forms of leishmaniasis are endemic in Algeria and highly endemic in East African countries, where outbreaks of visceral leishmaniasis are common. The epidemiology of cutaneous leishmaniasis in America is very complex, with variations in transmission cycles, reservoirs, sandfly vectors, clinical manifestations, and responses to treatment. In addition, there are several species of *Leishmania* in the same geographical area [69].

During its life cycle, *Leishmania* presents two stages: the promastigote, which is the infective flagellate form that develops in the digestive tract of the fly, and the amastigote, the parasite replicative form, in which the flagellum is diminished in size or absent. The life cycle begins when the female fly takes blood for feeding and ingests amastigotes present in a previously infected host (human or another mammal). Transformation to promastigote occurs within the next 24 to 48 h inside the insect vector. Once transformed, the parasite replicates in the intestine and migrates to the pharynx and esophagus. When the infected fly bites a new host, she inoculates between 10–100 promastigotes, which parasitizes the macrophages and dendritic cells, where they transform into amastigotes [69]. The diagnosis of visceral leishmaniasis is made by combining clinical examination, parasitological and serological tests, and PCR. The treatment of leishmaniasis depends on several factors, including the form of the disease, co-morbid conditions, the species of parasite, and geographic location. Treatment of visceral leishmaniasis consists of liposomal amphotericin B (AmB) or miltefosine, depending on the species of *Leishmania*. Alternatives include AmB deoxycholate and pentavalent antimonial compounds (sodium stibogluconate or meglumine antimoniate). A variety of topical and systemic treatments are available for cutaneous leishmaniasis.

Parasites require host iron for their survival. Indeed, the ability to efficiently acquire iron within a mammalian host is often an essential component of microbial virulence [70]. *Leishmania* spp., in both intracellular and extracellular forms, require iron for growth in vitro, which is often supplied in the form of hemin or other heme-containing compounds [71]. Promastigotes can rapidly take up ^59^Fe chelated by Lf, and this was inhibited by a 10-fold excess of unlabeled holo-Lf or apo-Lf [72]. Although iron is necessary for *Leishmania* growth, data from the same group have suggested that parasite-associated iron may also contribute to its susceptibility to death by macrophages, since iron acts as a catalyst for the formation of hydroxyl radicals ‘OH from hydrogen peroxide (H_2_O_2_) [73]. Later, the same group published that promastigotes of *L. chagasi* preferentially took up iron in a reduced rather than an oxidized form and can reduce iron from Fe^3+^ to Fe^2+^ prior to internalization. They found that both total and fractionated promastigotes exhibited NADPH-dependent iron reductase activity [74].

Silva et al. [75] assayed two antimicrobial peptides from the N1 domain of bLf, Lfcin17-30 and Lfampin265-284, as well as the hybrid version of Lfcin and Lfampin, (Lfchimera) against *L. donovani* promastigotes. All the peptides were leishmanicides, and Lfchimera showed significantly higher activity than its two constituent moieties. Lfchimera also showed activity against amastigotes of *L. pifanoi* in axenic cultures. When investigating the possible mechanism of action of the peptides, they found induced plasma membrane permeabilization and bioenergetic collapse of the parasites [75]. In a study carried out by Ashtana et al., Lf-appended AmB-bearing nanoreservoirs (LcfPGNP-AmB) were used for targeted eradication of *L. donovani* because macrophages have LfRs on their surface. LcfPGNP-AmB was architectured through ionic adsorption of Lf over core poly(d,l-lactide-co-glycolide) nanoparticles and characterized. LcfPGNP-AmB showed reduced toxicity, increased protective pro-inflammatory mediator expression, and downregulated disease-promoting cytokines. J774A.1 macrophages were infected with surface-adherent promastigotes to conduct an uptake study. For an in vivo study, Syrian hamsters were infected with *L. donovani* amastigotes for 30 days, and some hamsters were treated with a LcfPGNPAmB formulation. The results showed increased internalization of LcfPGNP compared with PGNP alone, and that LcfPGNPAmB delivered a high amount of the drug to the desired organ sites due to being an efficient APC-targeted drug delivery system. Targeted delivery directly reduced the AmB drug dose, which is highly desirable for an optimized therapeutic effect and diminished toxicity. In addition to these immunomodulatory effects, the potentiation of leishmanicidal activity by Lf favors the utility of developed nanoreservoir systems. Higher stability than commercial liposomal Ambisome and reduced toxicity than fungizone support the promising utility of Lf-appended nanoreservoirs as an alternative to problematic commercial formulations [76].

Recently, Halder et al. [77] reported the use of betulinic acid (BA), a pentacyclic triterpenoid from *Betula alba* bark. It was loaded onto uniformly spherical PLGA [poly (DL-lactide-co-glycolic acid) nanoparticles that carried BA on Lf (Lf-BANP). They found that the amastigote count in Balb/c mouse peritoneal macrophages was more effectively reduced by Lf-BANP than by BA or BANP without Lf. Lf-BANPs reduced the pro-parasitic, anti-inflammatory cytokine IL-10 but increased nitric oxide (NO) production in *L. donovani*-infected macrophages, indicating that LfBANP possesses significant anti-leishmanial activity. In addition, researchers have found that Lf-BANP has an immunomodulatory capacity [77].

In summary, bLf, its N-terminal derivatives, Lfchimera, and nanoparticles carrying Lf might be used alone or mixed with the drugs used to combat several leishmaniasis forms. In this way, the undesired effect of drugs and parasite resistance could be avoided.

### 3.6. Trypanosoma spp.

*Trypanosoma* is a notable genus of trypanosomatids, a monophyletic group of parasitic unicellular protists. Various species infect different vertebrates, including humans, causing trypanosomiasis diseases. Many species are transmitted by invertebrates, such as biting insects. The genus *Trypanosoma* consists of several dozens of species, and two of the three species that infect humans are pathogenic. In general, trypanosomatids are flagellated protozoa of the class Kinetoplastida, which go through different morphological stages (epimastigotes, amastigotes, and trypomastigotes) in their vertebrate and invertebrate hosts; however, the criterion for the three morphological stages has not been met for every species in the genus. For example, only *T. cruzi* and other species multiply in mammalian hosts as intracellular amastigotes, similar to those seen in the genus *Leishmania* [75]. In contrast, African trypanosomes, which cause sleeping sickness in humans and varying degrees of morbidity in domestic and wild mammals, do not have an intracellular form and multiply as trypomastigotes that circulate in the bloodstream and other extracellular spaces [75]. *T. cruzi* multiplication is discontinuous in the mammalian host, taking place at the amastigote stage. Development in the vector (Triatominae, or kissing bug) is completed in the large intestine, and mammals become infected by contaminant transmission. On the other hand, the *T. brucei* subspecies, *brucei,* multiplies continuously in mammals in the trypomastigote phase. Development in the vector (*Glossina* or tsetse fly) is completed in the salivary glands, and inoculative transmission to the mammalian host occurs. The two causative agents of human African sleeping sickness (African trypanosomiasis) are *T. brucei gambiense* and *T. brucei rhodesiense*. However, there are important differences in the transmission, pathogenesis, and clinical course of the two diseases and they have little in common except genetic and morphologic similarities in the etiologic agents.

### 3.7. Trypanosoma cruzi

Chagas disease, or American trypanosomiasis, is a life-threatening disease caused by *T. cruzi*. It is estimated that 6–7 million people are infected with this parasite worldwide. The disease is found mainly in endemic areas of 21 Latin American countries [69], where it is transmitted to humans and other mammals mainly through the feces or urine of blood-feeding triatomine insects. These insects live in cracks and crevices in the walls and roofs of houses and outdoor structures in rural and suburban areas.

Chagas disease has two phases. Initially, the acute phase lasts approximately two months after the infection is contracted. During this phase, a large number of parasites circulate through the bloodstream but, in most cases, there are no symptoms, or they are mild and non-specific. In less than 50% of people, a characteristic initial sign may be a skin lesion or a purplish swelling of an eyelid. In addition, fever, headache, enlarged lymph nodes, paleness, muscle aches, shortness of breath, swelling, and abdominal or chest pain can occur. During the chronic phase, the parasites remain hidden, mainly in the cardiac and digestive muscles. Up to 30% of patients have cardiac disorders, and up to 10% have digestive disorders (typically enlargement of the esophagus or colon), neurological disorders, or mixed disorders. Over the years, the infection can cause sudden death from cardiac arrhythmias or progressive heart failure. Chagas disease can be treated with benznidazole and with nifurtimox, which kill the parasite. Both drugs are almost 100% effective in curing the disease if administered early in the acute stage, even in cases of congenital transmission. However, the efficacy decreases as the time of infection elapses, and adverse reactions are more frequent in older ages. Treatment with these drugs is also indicated in cases of reactivation of the infection (for example, due to immunosuppression) and in patients at the beginning of the chronic phase, including girls and women of childbearing age (before or after pregnancy), to avoid congenital transmission of the infection [78].

*T. cruzi* can be internalized by mouse peritoneal macrophages (MPMs), human blood monocytes (HBMs), and human neutrophils and eosinophils. Furthermore, pre-treatment of MPM and HBM with Lf enhanced *T. cruzi* amastigote is associated with these cells. Immunofluorescence assay data indicate that HBM, MPM, and *T. cruzi* amastigotes bind Lf. Using ^125^I-labeled Lf, it was determined that each amastigote has an average of 1.1 × 10^6^ LfRs on the surface. Another interesting finding was that, in addition to Lf facilitation of amastigote internalization in HBM and MPM, Lf stimulates the killing of amastigotes by macrophage activation via oxygen reduction intermediates (H_2_O_2_, O^2−^, and ^1^O_2_) [79]. In another study, the same authors reported that apo-Lf and holo-Lf participate in the internalization of amastigotes in macrophages and result in activation that kills them via oxygen reduction intermediates [80,81]. In 1988, Lima et al. reported a specific surface marker for the *T. cruzi* amastigote form. They extended their observations and reported that the concentration of Lf that enhances phagocyte interaction with *T. cruzi* was within the range found in the plasma of patients with inflammatory conditions [78]. Moreover, Lima et al. reported a specific LfR for amastigotes, but this receptor is not found in promastigotes. The amastigote LfR may have biological significance in the parasite–host interaction since mononuclear phagocytes and MPM also express an LfR, and treatment of these cells with Lf has been shown to increase their capacities to take up and kill *T. cruzi* amastigotes in vitro [78].

### 3.8. Trypanosoma brucei

Human African trypanosomiasis is transmitted to humans through the bite of tsetse flies. Inhabitants of rural areas are most exposed to contact with the fly. The disease takes two forms, depending on the subspecies of the causative parasite: (1) *T. brucei gambiense* is found in 24 countries in western and central Africa. This form currently accounts for 97% of reported cases of sleeping sickness and causes chronic infection. A person can be infected for months or even years without presenting significant symptoms, which can appear when the disease is already advanced to the stage involving the central nervous system (CNS). In the first stage, trypanosomes multiply in subcutaneous tissues, blood, and lymph; the hemolymphatic phase is characterized by episodes of fever, headaches, lymphadenopathy, joint pain, and itching. In the second stage, the parasites cross the blood–brain barrier and infect the CNS. This is known as the neurological or meningoencephalic phase. Sleep cycle disorders, which give the disease its name, are an important feature of the second stage. This is usually when the most obvious signs and symptoms of the disease occur; and (2) *T. brucei rhodesiense* that is found in 13 countries in eastern and southern Africa. Today, this form accounts for less than 3% of reported cases and causes an acute infection. The first signs and symptoms are seen a few weeks to months after infection. The disease progresses rapidly and affects the CNS [82].

The disease can also be acquired via (1) mother-to-child transmission since trypanosomatids can cross the placenta and infect the fetus; (2) mechanical transmission through other hematophagous insects; (3) accidental pricks with contaminated needles; and (4) sexual contact. Screening for possible infection involves the use of serologic tests (only available for *T. brucei gambiense*) and physical examination for clinical signs, usually enlarged lymph nodes in the neck and, in some cases, analysis of cerebrospinal fluid. The type of treatment depends on the form and stage of the disease. The parasites can remain viable for long periods and reproduce to cause disease months after treatment. The drugs used in the treatment in the first stage are pentamidine, suramin, melalsoprol, eflornithine, and nifurtimox. Fexinidazole is used in both the first and second stages of the disease [82].

*T. brucei*, like other parasitic protozoa, needs iron to multiply and live. It has been described that this parasite uses Tf as a source of iron and binds to bLf through two LfRs of 40 and 43 kDa, respectively; apparently, these proteins are shared by hLf, hTf, and ovotransferrin. The N-terminal part of this receptor was identified as glyceraldehyde-3-phosphate dehydrogenase (GAPDH) [82].

### 3.9. Plasmodium spp. and Babesia spp.

Malaria is a disease caused by parasitic protozoa of the genus *Plasmodium: P. falciparum*, *P. vivax*, *P. ovale*, *P. malariae,* and *P. knowelsi*; *P. falciparum* is the most lethal species. Malaria infects 243 million people and causes 1.5 million deaths each year (affecting mainly children and pregnant women) [83]. Plasmodium has a very complex life cycle, with biochemical and physiological adaptations depending on its host environment. Merozoites in the vertebrate host and sporozoites in the mosquito salivary glands are devoted to host cell invasion, while liver- and blood-stage schizonts and oocysts are focused on replication and cell division. The life cycle requires extensive transcriptional regulation of the *Plasmodium* genome [84]. Sporozoites injected into a host by an infected mosquito migrate to the liver and initiate the incursion of hepatocytes. Inside hepatocytes, they multiply and differentiate into schizonts containing thousands of hepatic merozoites. These are released into the blood, initiating the erythrocytic stage when they invade and replicate within the RBCs. Asexual blood parasites differentiate into gametocytes that ensure parasite transmission to the mosquito [85].

The beneficial effect of milk against *Plasmodium* was first observed 70 years ago in monkeys infected with malaria parasites [41]. Animals infected with *P. cynomolge* by blood inoculation developed poor parasitemia. The suppression reaction was observed while animals were on a milk diet, but days after this diet was changed, the reaction ended, and a critical infection was confirmed. Other observations of African people who normally present iron deficiency showed apparent suppression of malaria [40]. Regarding the importance of iron in the outcome of infections, many microorganisms show enhanced virulence when the iron concentration is increased and their growth is suppressed when iron is reduced by iron chelators; thus, iron chelators were tested in vitro against cultures of *P. falciparum*. The results show that both iron-free and iron-saturated human Lf inhibited the growth of this microorganism. This inhibition depended on iron deprivation as well as on the generation of oxygen-free radicals that damaged the membrane of both the parasite and RBCs [86]. In human malaria, Lf derived from neutrophils acts by lowering the amount of iron available to the parasites in serum, but the destructive effect on the membranes by oxygen-free radicals could be more important [86].

In the host, malaria sporozoites colonize hepatocytes, where they develop in large numbers of RBC-infective merozoites. The sporozoite surface is covered by the circumsporozoite protein, which mediates the selective invasion of hepatocytes by binding to heparan sulfate proteoglycans (HSPGs) and receptor–ligand interactions [87,88]. Lf interferes with the invasion and colonization of hepatic cells by sporozoites. Malaria sporozoites depend on low-density lipoprotein (LDL) receptor-related protein (LRP) and on cell surface HSPGs for host cell invasion through chondroitin sulfate A (CSA) binding to LRP at the same site that apolipoprotein E (apo E) and Lf bind [87,88]. Even though HSPGs are widely distributed in animal tissues, it is interesting that sporozoites, Lf, and apo E (which mediates dietary lipid transport toward the liver and is rapidly removed from the blood circulation by the liver), are retained almost exclusively in the liver. In such a manner, Lf interferes with and avoids the invasion and colonization of hepatic cells by sporozoites. Stunningly, the Lf and lipoprotein clearance pathways of the host in malaria sporozoite invasion of the liver compete in vitro and in vivo for the same binding sites as the CSA protein of malaria sporozoites. This could explain the lower parasite densities and fewer episodes of clinical malaria observed in breast-fed neonates feeding breast milk with high concentrations of Lf and fat [88]. Cell surface HSPGs facilitate the LRP-mediated endocytosis of Lf and apo E.

Later, the effects of Lf on the adhesion of RBCs infected with *P. falciparum* (PE) were studied because Lf binds to CSA, which is the receptor for PE. It has been observed that Lf inhibits adhesion to other endothelial cell receptors, such as CD36, CD36-CHO cells, and thrombospondin. This inhibition is due to Lf and PE sharing the same binding site. It is possible to exploit this multifunctional property by designing an anti-adhesion peptide from amino acid residues 25–37 of Lf. This peptide could be used as a template to develop anti-adhesion agents that avoid *P. falciparum* invasion [89].

On the other hand, epidemiological and laboratory studies have demonstrated the positive contributions of breastfeeding in reducing the incidence and severity of intestinal and respiratory infections. Thus, the presence of anti-malaria antibodies and the inhibitory component in human milk could affect parasite growth. Several specific anti-*P. falciparum* antibodies for all intra-erythrocytic states were found in Nigerian maternal and infant sera, as well as in Nigerian milk samples. sIgA is the predominant antimalarial antibody in breastmilk, suggesting an important role in milk-derived immunity against malaria. The in vitro inhibition of parasite growth by breastmilk from Nigerian mothers was 61%, while that of breastmilk from US mothers was 51%. These milk samples did not contain any measurable concentrations of anti-parasite antibodies. Thus, the level of inhibition suggests the role of non-antibody factors, such as Lf, in the in vitro inhibition of parasitic growth. Lf inhibited 78% and 79% of growth when combined with sIgA compared to 62% with the sIgA fraction alone. The growth inhibition obtained from combining sIgA and Lf was not significantly different from that of Lf alone, although it suggests a major role of Lf in the inhibition process. The authors found evidence of the dual activity of Lf mentioned above. Thus, the level of inhibition suggests the role of non-antibody factors such as Lf in the in vitro inhibition of parasitic growth. In the presence of Fe^3+^, Lf increases the production of oxygen radicals, which damage the parasite membranes and increase Pes; as an iron-binding protein, it deprives microbial organisms of the free iron required for their metabolic activities. It was also proposed that Lf could interfere with the iron-containing metabolic enzymes of malaria parasites, in addition to the high concentration found in Nigerian and US milk [49,90].

Among the antimicrobial peptides that provide protection against unrelated pathogens, an increase in Lf concentration in BCG immunization has been observed to prevent adult pulmonary tuberculosis, as in malaria infections, where vaccination partially protects against infection. According to this observation, the authors proposed treating malaria with Lf and a cathelicidin-type peptide, and an important reduction in the parasitemia levels in infected mice was found [90]. Commonly, malaria infections are treated with quinine compounds that are effective, but parasite-acquired resistance has been found. These results were considered to investigate the use of an antimalarial combination therapy that aims to attack different characteristics of the parasite in a synergic form to obtain a better treatment and evaluate the effect in an in vivo malaria model [91]. The effect of buffalo Lf (buLf) was studied because it has more antimalarial activity than Lf of bovine origin. Significantly, this was the first report of the antimalarial effect of buLf encapsulated in alginate-chitosan-calcium phosphate nanocapsules (NCs). This protein is more effective inside NCs, and helped mice survive up to 35 days post-infection, compared with the control mice that died at 15 days post-infection. The biodistribution of buLf-NCs showed increased concentration in various vital organs, demonstrating the better release and protection from degradation in the gastrointestinal tract. In addition, Lf protected mice from anemia, maintaining an iron concentration adequate for its metabolism, as well as the adaptive immune response at protecting levels. Nanoformulated buLf is proposed as both a therapeutic agent and for combination therapy with commonly used drugs, the side effects of which can be overcome with the help of Lf [91].

Research on novel or improved means to control malaria by natural proteins is important to avoid parasite resistance and the adverse effects of drugs in the host. Lf is an important candidate for malaria treatment. Lf can damage malaria parasites in at least three ways: (1) through iron chelation, thereby preventing the parasite from obtaining it; (2) production of oxygen radicals that affect the parasite membrane; and (3) competing with the infected cells for the receptor in the host cell.

On the other hand, *Babesia* is a genus of intra-erythrocytic protozoan parasites belonging to the exclusively parasitic phylum, Apicomplexa. There are more than 100 known species of this genus, mainly in mammals, but also in birds, and all transmitted by ticks, which are blood-sucking arthropods. In general, ixodid (hard-bodied) ticks are vectors for *Babesia* spp., but a small number are transmitted by argasid (soft-bodied) ticks. For a long time, it was thought that *Babesia* spp. only affect domestic animals, but in 1957, babesiosis was described in a human for the first time when the disease presented a fulminant and ultimately fatal infection in a farmer in Croatia, so it became a high-impact zoonotic disease [92,93].

The life cycle of *Babesia* begins when sporozoites are transmitted from the infected tick to the mammalian host, where they infect erythrocytes and begin to replicate asexually. They undergo repeated rounds of replication, exit, and invasion, resulting in hemolysis. A subset of parasites will perform the formation of gametocytes, the stage transmitted to the tick. Once ingested in a blood meal, gametocytes differentiate into gametes, which are released from the RBCs into the midgut. Two gametes combine to form a diploid zygote (the only stage of the diploid life cycle), which then invades the epithelial cells of the tick intestine and undergoes meiotic division to form haploid kinetes. These invade various tick tissues and replicate asexually, including the ovaries (except *B. microti*, which mainly invades kidney cells), which will allow them to be transmitted to tick larvae. The kinetes that form in the cells of the tick tissue migrate to the salivary glands. In these, the parasite forms a dormant sporoblast that only begins to replicate into sporozoites capable of being transmitted to the mammalian host, when the tick begins to feed on blood [94]. The main pathological manifestation of *Babesia* infection is the destruction of erythrocytes, leading to hemolytic anemia with complications due to the release of toxins and waste products into the vascular bed. Cytokine storms can cause further damage as the host immune system responds to the infection [93].

In 2005, the effect of Lf on *B. caballi* and *B. equi* (reclassified as *Theileria equi*) was investigated. These pathogens infect the erythrocytes of horses and induce fever, edema, anemia, and jaundice. The disease causes important economic losses in the equine industry. In this trial, the authors infected erythrocytes for 24 h with *Babesia* to achieve parasitaemia. After 24 h, the medium was changed by adding bovine native-Lf (∼38% Fe^3+^), bovine Lf hydrolyzate, iron-saturated holo-Lf (∼70% Fe^3+^), and apo-Lf (0% Fe^3+^). The tests were carried out for 4 days. They showed that *Babesia caballi* is only suppressed by bovine apo-Lf at concentrations of 2.5 and 5 mg/mL. *Babesia equi* was not affected by any treatment. They tested the chelation capacity of apo-Lf in RPMI medium, which was used for the assay and found no levels of iron bound to Lf; thus, they decided that the inhibitory effect is not due to the iron binding of the medium. This finding is interesting since the authors suggest that, according to their results, it could be a new mechanism of inhibition towards microorganisms, specifically towards *Babesia*, perhaps focusing on a suppression of some growth factor specifically necessary for *B. caballi* [95].

### 3.10. Toxoplasma gondii

Toxoplasmosis is a disease caused by *Toxoplasma gondii*, one of the most common parasites in the world. More than 60 million people in the United States are infected with this parasite. Toxoplasmosis is a zoonosis with a worldwide distribution. Invertebrates such as flies and cockroaches can contribute to the spread of oocysts, which accompany cat defecation. The fecal oocysts are not infective; they must go through a differentiation process in the soil that lasts up to three weeks and can remain infectious for a long time in moist soil [96] or through mother-to-child transmission during pregnancy. *T. gondii* is an intracellular parasite that resides in parasitophorous vacuoles, thus avoiding fusion with lysosomes [97]. *T. gondii* is also capable of invading macrophages.

*T. gondii* has a complex life cycle in which humans and other animals participate as intermediate hosts, with cats and other felines being the definitive hosts. The parasite occurs in three different forms: tachyzoites (a trophozoite mainly found in the brain and muscles), tissue cysts, and oocysts. The latter are only produced in the intestines of the definitive host. Primary infection is an acute phase of the disease where the parasite divides rapidly (tachyzoites) and triggers the activation of the immune system, which, if it is effective, will control the infection with the consequent formation of cysts that contain parasites dividing slowly (bradyzoites); this is the chronic phase. HIV-positive people and pregnant women are particularly at risk of contracting the disease. Toxoplasmosis is diagnosed by isolating the parasite through the inoculation of laboratory animals or cell culture or with a serological profile, which may not be reliable in immunocompromised patients or in a fetus [98]. However, there are tests capable of detecting serological or urine antibodies created by the immune system to fight the parasite, especially an increase in IgG levels and/or the presence of IgM-specific antibodies [99].

*T. gondii* is sensitive to pyrimethamine and sulfonamides, which are used in a combination treatment that increases their effect more than six times of that individually [100]. Similar to other cells, *T. gondii* needs iron to develop its metabolic functions and reproduce. Bovine Lf can activate macrophages to phagocytize and inhibit the growth of various microorganisms and even kill them [9,101]. In in vitro experiments with mouse peritoneal macrophages (MPMs) and *T. gondii*, Tanaka et al. [102] measured the inhibition of intracellular parasite growth in the presence of bLf. They first observed that the presence of bLf did not result in any type of damage to host cells. When parasites were pre-treated with bLf, they did not inhibit macrophage entry or growth within these cells, indicating that Lf had no parasiticidal effect on *T. gondii* [102]. To determine the anti-toxoplasma effect of bLf in MPM, the researchers incorporated 3H-uracil in MPM infected for different times. They found that when supplemented with a Lf concentration of 1000 µg/mL, 3H-uracil incorporation was 27.8% compared to the control without Lf. They also found that MPM produces an inhibitory effect on the intracellular parasite in the presence of bLf in the culture and requires the existence of Lf to maintain the activity. Furthermore, this inhibitory effect was obtained in the MPM group treated with apo-Lf or holo-Lf. Thus, the inhibition of intracellular parasite growth is independent of Fe [102]. In another study, the authors explored the mechanisms of the inhibitory activity induced by Lf. It is known that murine macrophages activated with IFN-ɣ and/or LPS display killing activity against *T. gondii*, and this activity is associated with increased production of nitrogen oxide (NO). To clarify the effector pathway of *T. gondii* growth inhibition induced by Lf in macrophages, the authors examined the production of free radical oxygen products, O^2−^ and NO, in murine macrophages stimulated with Lf. To evaluate the role of NO derived from L-arginine in the mechanism of this activity, the culture medium was supplemented with a competitive inhibitor of the L-arginine-dependent effector pathway, NG-monomethyl-L-arginine acetate (NGMMA) and the viability of the intracellular parasites in mouse macrophages was monitored [103]. The production of O^2−^ diminished in cultures of macrophages supplemented with Lf, and the effect of Lf was dose- and time-dependent. On the other hand, the production of NO was enhanced in cultures of macrophages supplemented with IFɣ but not with Lf. These findings suggest that the growth–inhibitory activity induced by Lf in macrophages is not mediated by O^2−^ or NO molecules. When the competitive inhibitor, NGMMA, was used, the inhibitory effects induced by IFɣ were virtually abolished. Similarly, the inhibitory activity induced by Lf was diminished in cultures supplemented with NGMMA. From these findings, the authors concluded that an L-arginine-dependent effector pathway that does not involve NO production might mediate the inhibition of *T. gondii* growth induced by Lf in macrophages. More recently, Dzitko et al. [104] showed that human Lf inhibited the intracellular growth of *T. gondii* tachyzoites.

Regarding the effect of Lf-derived peptides on *T. gondii* parasites, Tanaka et al. (1996) reported the parasiticidal effect of Lfcin B. They incubated *T. gondii* with Lfcin B at 100 or 1000 µg/mL, with bLf at 1000 µg/mL, or with a peptide of the C-terminal region of bLf at 1000 µg/mL and found that parasites treated with 100 µg/mL of Lfcin B for 1 h had 64% mortality. When *T. gondii* was treated with 1000 µg/mL of Lfcin B, they reached 96% mortality in 0.5 h, in contrast to 80% viability with parasites treated with bLf or with the Lf-C-terminal peptide. The authors also assayed the penetration ability of *T. gondii* into mouse embryonal cells (MECs) after pre-incubation with Lfcin B at 1000 µg/mL for 0.5 h and observed significantly less penetration activity. Meanwhile, the parasites treated with 100 µg/mL showed gradually increasing penetration activity, and only 10% of the cells were infected after 4 h of treatment [105]. Isamida et al. [106] extended the previous observation of Lfcin B by performing an experimental infection of mice with *T. gondii*. Thirty mice were orally challenged with *T. gondii* cysts at a dose of LD50 or LD90 and divided into three groups. Thirty minutes before challenge and on each of the first 7 days pos-tinfection, 0.5 or 5 mg of Lfcin were orally administered in 0.5 mL of saline solution. Another two groups of five mice were intraperitoneally administered either 0.1 or 1 mg of Lfcin in 0.5 mL of saline solution during the first 7 days post-infection. As a negative control, five mice were orally administered saline solution. To study different routes of infection, five mice were orally administered 5 mg of Lfcin and challenged with intraperitoneal *T. gondii* cysts at a dose of LD50. After the challenge, the mouse mortality and survival were monitored for 35 days post-challenge. On day 35, the mice were sacrificed, and the brain of each mouse was removed and homogenized in PBS to count the number of cysts. They found that the mice orally inoculated with *T. gondii* cysts and orally administered 5 mg of Lfcin all survived at least 35 days post-challenge. However, 60% of the mice orally administered 0.5 mg of Lfcin died of acute toxoplasmosis 14 days post-challenge. On the other hand, a substantial number of tachyzoites were recovered in the peritoneal cavity of mice administered 1 mg of Lfcin intraperitoneally, and 40% of the mice died of toxoplasmosis within 21 to 31 days post-challenge. In the control group, 60% of the mice administered only saline solution died of acute toxoplasmosis within 14 days post-challenge. In the case of mice orally administered a unique LD50 dose of cysts, 20% of the mice died of toxoplasmosis within 7 days post-challenge. In the mice orally administered with 5 mg of Lfcin, the number of cysts/mouse brain was significantly lower than that of the survivors in the control group. The number of cysts in the survivors in the group orally administered with 0.5 mg of Lfcin was lower than that in the control. When 1 mg of Lfcin was intraperitoneally administered to mice, individual differences in the number of cysts were found, and the average number of cysts was similar to that of the control [106]. Omata et al. [35] found that *T. gondii* sporozoites pre-incubated with Lfcin showed decreased activity in the penetration of MECs. Therefore, mice inoculated with 10^5^ sporozoites pre-incubated with Lfcin showed a higher survival rate than those inoculated with the same number of sporozoites without Lfcin. Using fluorescence and far western blot assays, Tanaka et al. found an Lf, Tf, and oTf binding protein in *T. gondii* that is 42 kDa and may participate in iron uptake by the parasite [82].

### 3.11. Trichomonas vaginalis

*Trichomonas vaginalis* is a flagellated parasitic protozoan that causes trichomoniasis, a disease of the genitourinary system. Trichomoniasis is one of the most common causes of urogenital sexually-transmitted diseases of non-viral etiology in humans, with a global prevalence rate of 170 million cases per year. According to data from the WHO, it represents almost half of all curable sexually-transmitted diseases [107]. *T. vaginalis* varies in size and shape, and ameboid forms have been observed in parasites adhered to the vaginal mucosa. In pure cultures, the classic form of this pathogen is pear-shaped. *T. vaginalis* does not have a cystic phase and only exists as a trophozoite (trichomonads). Trichomonads divide by binary fission and give rise to progeny in the lumen or tissue surfaces of the human genitourinary tract [108]. Trichomonads can survive for a long time in the acidic environment of the vagina, and the disease can become chronic. Clinical manifestations in women range from an asymptomatic carrier state in approximately 50% of infected women to symptomatic patients that may have mild to severe inflammation with foul-smelling exudate and severe irritation. Infection during pregnancy may be associated with premature rupture of the membranes, pre-term delivery, and low birth weight babies. In men, clinical manifestations range from an asymptomatic carrier state to an acute state characterized by suppurative urethritis, dysuria, and mild pruritus [109]. The first manuscript in which the interaction of Lf with *T. vaginalis* is mentioned is that of Peterson and Alderete in 1984 [110]. The objective was to demonstrate that one of the main mechanisms of the pathogenesis of parasites, specifically *T. vaginalis*, is to obtain iron from iron-containing proteins through a receptor. Iron is generally associated with different plasma proteins, such as Tf and Lf. The results of this study were the discovery of two high-affinity specific receptors with molecular weights of 75 and 178 kDa, both stable at temperatures of 4 and 37 °C. Interestingly, an increase in intracellular iron and higher activity of pyruvate-ferredoxin oxidoreductase were found, implying that iron is important for parasitic intracellular metabolism [110]. The Lf used in this research was purified from human milk and saturated with 1 µg of FeCl_3_/1 mg protein. These results contribute significantly to explaining the pathogenesis of the disease since it is only known that *T. vaginalis* results in hemolysis, and the subsequent events after this phenomenon are unknown. Like hemoglobin, Lf is an iron-binding protein, and *T. vaginalis* has receptors for hemoglobin that also allow for it to take up iron. Similarly, studies with different levels of iron supplementation in the culture medium of *T. vaginalis* and its interaction with Lf were reported by Lehker and Alderete [111] in 1992. They observed that under conditions of iron deprivation in the medium, receptors for Lf increased and that these receptors decreased when iron was added to the medium.

Likewise, in studies with media with a low amount of iron, when Lf saturated with iron was added, receptors for Lf were also reported to be elevated. The results also indicate that there are preformed receptors that can be mobilized when *T. vaginalis* is in a medium with a low amount of iron, in addition to expressing new LfRs. Interestingly, proteins with molecular weights of 136 and 72 kDa were found to be expressed only at iron levels ≤ 50 µM, while a protein with a molecular weight of 128 kDa was only expressed after growth in iron concentrations ≥ 50 µM. These studies demonstrate that iron is key in the pathophysiology of the disease and that different levels of iron in the environment where *T. vaginalis* is living can trigger intracellular signaling cascades or function as a transcription factor for the pathogen.

Recently, the effect of Lf on *Trichomonas gallinae*, a pathogen that causes disease in birds, was assayed. In vivo and in vitro analyses were carried out with the nanoformulation of different concentrations of Lf and metronidazole (nano-MLf). The in vitro assay viability was tested with an inoculum of 10^4^ trophozoites for 12, 24, and 48 h. The in vivo test was performed on pigeons inoculated with 30,000 viable trophozoites; this test was carried out for 5 days, and the nanoformulation was given orally once a day. They showed that with the highest dose tested (100 ug/mL nano-MLf), there was 100% eradication of trophozoites from 12 h of treatment. Doses of 12.5 ug showed complete eradication after 24 h, and with doses of 1.5 ug, no viability was found at 48 h. On the other hand, in the in vivo assay, no viable trophozoites were observed in the crop lavage from the first dose of nano-MLf (50 mg/kg). Interestingly, by day 5 of treatment, all animals challenged with the treatment recovered from the disease [112].

The authors compared this formulation with metronidazole and observed better and earlier eradication when using the nano-MLf. The results suggest that this nanoformulation could be a new alternative for the generation of drugs in birds [112]. This report is of great interest since it is the first study in which the effect of Lf on *Trichomonas* is observed. In addition, membrane receptors have been reported in *T. vaginalis* that are possibly present in *T. gallinae* and that are responsible for the microbiocidal effect of Lf towards this pathogen (Figure 2) (Table 1).

**Table 1 pharmaceutics-14-01702-t001:** Effect of Lactoferrin and its peptides on diseases caused by pathogenic protozoa.

Disease/Pathogen	Treatment	Effect	Reference
Cryptosporidiosis/*Cryptosporidium parvum*	hLf, hLf hydrolyzed, hLfcin B 10 µg/mL	Lf hydrolyzed and Lfcin B affected the viability, showed parasiticidal effect, and decreased the infectivity. Lf had no effect.	Murdock, C.A. and K.R. Matthews, 2002 [27]
	Lf 10 mg/mL	Lf affected the viability, showed parasiticidal effect, and decreased the infectivity. No significant effect was observed on the oocyst phase or parasite intracellular development.	Paredes, J.L., et al., 2017 [29]
Eimeriosis/ *Eimeria stiedai*	bLfcin B 1000 ug/mL (In vivo rabbits and mice)	Less infectivity and less penetration into host cells, lower number of oocysts, the liver had fewer abscesses, and they did not present inflammation of the bile ducts.	Omata, Y., et al., 2001 [35]
Amoebiasis/ *Entamoeba histolytica*	hLf, hLfcin B, bLf, bLfcin B 100 µM Metronidazole 58.4 µM	The amoebicidal effect and the synergistic amoebicidal effect was found between metronidazole and Lf or metronidazole and Lfcin.	Leon-Sicairos, N., et al., 2006 [46]
	hLf, sIgA, Lysozyme 100 µM	The amoebicidal effect, rearrangements, and disruption in the lipid pattern after being bound to the amoeba membrane.	Leon-Sicairos, N., et al., 2006 [51]
	Lfcin 17–30, Lfampin 265–284, Lfchimera 100 µM	Amoebicidal effect, Lfchimera showed the strongest amoebicidal activity.	Lopez-Soto, F., et al., 2010 [53]
	bLf 20 mg/kg (In vivo C3H/HeJ mice)	Elimination of amoebiasis by the production of anti-amoeba IgA antibodies and amoebicidal effect.	León-Sicairos, N., et al., 2012 [56]
	bLf 2.5 mg/100 g Metronidazole 0.5 mg/100 g (In vivo Syrian hamster)	No clinical signs of disease and amoebiasis was effectively decreased. Liver function and blood cells approached normal levels.	Ordaz-Pichardo, C., et al., 2012 [58]
	bLfcin-B, bLfcin 17–30, bLfampin 265–284, 250, 1000 µM bLfcin-B, bLfcin 17–30, bLfampin 265–284 10 mg/kg (In vivo C3H/HeJ mice)	Concentrations higher than 250 µM of Lfampin showed necrosis. No effect of Lfcin-B or Lfcin 17–30. Mice in the groups treated with Lfcin 17–30 or Lfcin B showed an absence of amoebic trophozoites in the intestinal lumen on 75% of the animals and treatment with Lfampin eradicated the infection in 100%.	Diaz-Godinez, C., et al., 2019 [59]
Giardiasis/ *Giardia duodenalis*	hLf, 2.5 mg/mL bLf, 2.0 mg/mL hLfcin 18–40, 24 µg/mL bLfcin 17–41, 12 µg/mL.	bLfcin had the most potent giardicidal activity. Log-phase cells were more resistant to killing than stationary-phase cells	Turchany, J.M., 1995 [63]
	bLf, 12.5 μM bLfcin 2.6 μM	Trophozoites exhibited structural changes in membranes, cell growth cessation, and production of immature cysts.	Frontera, L.S., et al., 2018 [60]
	bLfcin17-30, bLfampin265-284, Lfchimera 40 µM metronidazole 100 µM albendazole 5 µM	Parasiticidal effect. Apoptosis of trophozoites. Lfampin showed the best microbicidal activity. Lf and its derivative peptides showed synergy with the drugs.	Aguilar-Díaz, H., et al., 2017 [65,66]
	Lf, maltodextrin 0.5 g (in vivo, children aged 12–36 months)	No significant difference between the two groups in symptomatology. There was a lower prevalence of colonization with *Giardia* spp. and better growth in children of the group treated with Lf	Ochoa, T.J., et al., 2008 [67]
Leishmaniasis/ *Leishmania donovani*	Lfcin 17–30, Lfampin 265–284, Lfchimera 10 µM	LFchimera was the most active peptide. All peptides induced plasma membrane permeabilization and bioenergetic collapse of the parasites	Silva, T., et al., 2012 [75]
	Nanoparticles of Lf and amphotericin B. (In vivo Syrian hamsters) 0.8 µg/mL	Leishmanicidal effect. Increased protective of proinflammatory mediators expression and down-regulation of disease-promoting cytokines.	Asthana, S., et al., 2015 [76]
	Nanoparticles of Lf with betulinic acid (In vivo Balb/c mouse) 1.5 mg/mL	Leishmanicidal effect. Reduced anti-inflammatory cytokine IL-10 and increased nitric oxide.	Halder, A., et al., 2018 [77]
Chagas disease/*Trypanosoma cruzi*	hLf 10 µg/mL	Microbicidal effect. Lf stimulated the killing of amastigotes by macrophage activation via oxygen reduction intermediates.	Lima, M.F. and F. Kierszenbaum, 1985 [78]
	hApo-Lf hHolo-Lf 10 µg/mL	Microbicidal effect. Lf participated in the internalization of amastigotes in macrophages and stimulated respiratory burst.	Lima, M. F. and. Kierszenbaum, F. 1987 [79]
Malaria/ *Plasmodium falciparum*	hApo-Lf hHolo-Lf 30 µM	Inhibition of *Plasmodium* depended on iron deprivation as well as on the generation of oxygen-free radicals that damaged the membrane.	Fritsch, G., et al., 1987 [78]
	Heparinase 10 mU/mL, anti-LRP 10 µg/mL, GST-RAP 150 µg/mL, Lf 400 µg/mL	They all cause inhibition. Only Lf interfered with sporozoite attachment by binding to both receptor-related proteins and heparan sulfate proteoglycans.	Shakibaei, M. and U. Frevert, 1996 [87]
	Lf, Tf 1.6 µM ApoE, β3-VLDL apoE-enriched with 3-VLDL 250 µg/mL	Lf and lipoproteins inhibited the binding to liver cells by competition for the same binding sites as the CSA protein of sporozoites. No effect of 3-VLDL.	Sinnis, P., et al., 1996 [88]
	hLf, Lfcin 25–37 100 μg/mL KPSE peptide 394–409 of CD36 FASP peptide; 300–312 of CD36	The inhibitory effects of Lf on *P. falciparum*–infected erythrocytes binding was a result of specific binding of Lf to CD36 and thrombospondin. No effect of KPSE peptide and FASP peptide.	Eda, S., et al., 1999 [89]
	Lf 50 µg/mL cathelicidin LL-37 peptide 50 µg/mL (in vivo naïve mice)	A 95% reduction in malaria infection was observed after treatment with Lf and 43% with cathelicidin.	Parra, M., et al., 2013 [90]
Malaria/ *Plasmodium berghei*	BuLf, Nanoformulation with BuLf Chloroquine (in vivo, BALB/c mice)	BuLf and Nanoformulation with Lf showed lower parasitaemia, low inflammation in the spleen, free radical ion production, and higher survival tendency compared to the chloroquine group.	Anand N, et al., 2015 [91]
Babesiosis/ *Babesia equi (Theileria equi) Babesia caballi.*	Native-bLf bLf hydrolysate, bHolo-Lf bApo-Lf 2.5 and 5 mg/mL	Only Apo-Lf suppressed *B. caballi*. No effect on *B. equi*.	Ikadai, H., et al., 2005 [95]
Toxoplasmosis/*Toxoplasma gondii*	bLf, Lfcin B Lf C-terminal region 1000 µg/mL	Lfcin B produced an inhibitory effect and loss infectivity of parasites.	Tanaka, T., et al., 1995 [102]
	Lfcin B, bLf Lf C-terminal region 1000 µg/mL	Parasiticidal effect and less penetration activity. Lfcin B was shown to be more effective than the other treatments.	Tanaka, T., et al., 1996 [103]
	Lfcin B 5 mg (In vivo, mice)	Lfcin orally administered caused survival and reduced the number of cysts in the brains of mice compared to the control group.	Isamida, T., et al., 1998 [106]
Trichomoniasis/ *Trichomonas gallinae*	Nanoparticles with Lf and metronidazole, metronidazole 100 ug/mL (In vivo, pigeons 50 mg/kg)	100% eradication of trophozoites with the nanoformulation. Pigeons treated with nanoparticles showed non-viable trophozoites in the crop lavage and all challenged animals recovered from the disease.	Tabari, M.A., et al., 2021 [112]
Primary amoebic meningoencephalitis/ *Naegleria fowleri*	Tritrypticin, Lf, killer decapeptide, scrambled peptide 100 µg/mL	Only tritrypticin showed a positive effect causing marked apoptosis.	Tiewcharoen, S., et al., 2014 [113]
Amoebic keratitis/ *Acanthamoeba castellanii*	Lf 1.48 mg/mL	No statistically significant difference was observed compared to the control treatment.	Alsam, S., et al., 2008 [114]
	Apo-bLf, holo-bLf, native bLf 10 µM	Apo-bLf affected the viability of the parasite, caused trophozoite death and did not cause encystment. No effect with holo-bLf or native bLf.	Tomita et al. in 2017 [115]

Lactoferrin (Lf), Buffalo Lactoferrin (BuLf), Human Lactoferrin (hLf), bovine Lactoferrin (bLf), Human Lactoferricin (hLfcin), bovine Lactoferrcin (bLfcin), Lactoferricin B 4–14 (Lfcin B), Apo-lactoferrin (Apo-Lf or only Lf), Bovine apo-lactoferrin (Apo-bLf or only bLf), Human apo-lactoferrin (Apo-hLf or hLf), Bovine holo-lactoferrin (bHolo-Lf), Human holo-lactoferrin (hHolo-Lf), Lactoferrampin 265–284 (Lfampin), Bovine Lactoferrampin 265–284 (bLfampin), Lactoferrin chimera consists of Lfcin 17–30 and Lfampin 265–284 (Lfchimera).

## 4. Effect of Lactoferrin on Free-Living Pathogenic Protozoa

Free-living amoebas (FLAs) are eukaryotic and mitochondrial microorganisms that can carry out their life cycle as parasites or inhabit natural environments as FLAs. Given this, FLAs are categorized as amphizoic. These microorganisms are a group of opportunistic protozoa that cause serious health problems in humans and animals. Furthermore, amoebas can be vectors of pathogenic bacteria [116].

### 4.1. Naegleria fowleri

Different microorganisms can cause disease in different systems. In the central nervous system, an immunoprivileged organ, pathogens need to cross the blood–brain barrier to cause damage and establish themselves in the tissue. With their dissemination throughout the body, pathogens also encounter proteins of the innate immune system, such as Lf [117]. Among the pathogens, *Naegleria fowleri* is the main amoeba capable of causing disease in brain tissue.

*N. fowleri* is an FLA that enters the nasal mucosa and colonizes the brain, causing primary amoebic meningoencephalitis (PAM) in humans. Various virulence mechanisms, such as adhesion, locomotion, phagocytosis, and secretion of proteases that degrade host proteins, have been reported [118,119,120]. Clinically, *N. fowleri* causes an acute and fulminant disease that can lead to death 7–10 days after the pathogen enters the body. This disease commonly occurs in young immunocompetent patients, especially after contact with water contaminated with amoebas. Although the disease is extremely rare, it has a mortality rate of 98%. Establishing an early diagnosis is essential for the survival of the patient. The acute course, misinformation, and non-specific symptoms hinder the development of successful treatments and effective diagnostic tools. In addition, this disease is generally not reported; typically unrecognized, it is often confused with bacterial or viral infections and is therefore underdiagnosed [120].

Antimicrobial peptides are important components of the host innate immune system against diverse pathogenic etiologies. In this sense, one study has reported the interaction of Lf with the *N. fowleri* strain Siriraj. In this work, several antimicrobial peptides were tested: tritrypticin, Lf, the killer decapeptide, and the scrambled peptide at doses of 100 µg/mL in *N. fowleri* trophozoites (2 × 10^5^ cells/mL) for 30 min and 1, 3, 6, and 12 h. According to their results, only tritrypticin had a positive effect on *N. fowleri*, causing marked apoptosis at the incubation times used that was comparable to the positive control (amphotericin B) [113]. Although they did not find any effect of Lf on *N. fowleri*, it is important to note that the dose was very low compared to other studies [12]. In addition, the sequence shown for Lf is a short sequence of Lfcin B, which is not mentioned, even though this peptide is reported to have a better antimicrobial effect than the native protein [121]. Moreover, the effects of Lf and other antimicrobial peptides toward the virulence factors of *N. fowleri* could be studied. It is important to consider these aspects and perform other tests with different variables to establish a final decision regarding the effect of Lf on *N. fowleri*.

### 4.2. Acanthamoeba castellanii

Another pathogen that is included within the group of FLAs is *Acanthamoeba castellanii*. *Acanthamoeba* spp. are microorganisms that are distributed ubiquitously throughout the environment. This amoeba has two phases in its life cycle: an active trophozoite phase that exhibits vegetative growth and a latent cyst stage with reduced metabolic activity. *Acanthamoeba* may have the ability to be an opportunistic pathogen: it causes granulomatous amoebic encephalitis (GAE) in immunocompromised patients or can cause infection in immunocompetent patients through different clinical presentations, such as skin lesions, sinusoidal infections, or, more commonly, amoebic keratitis (AK) [122]. Although AK/GAE morbidity data differ around the world (0.13 to 33 cases per million), the numbers have been steadily increasing in recent years, particularly in developed countries. Since GAE occurs in immunocompromised patients, the mortality figures are very high, ranging between 95–98% among infected patients. This high mortality rate is due to several factors—as with *N. fowleri*—including late diagnosis or misdiagnosis of infection and/or lack of effective therapeutic agents, especially against resistant cystic forms of *Acanthamoeba* [123].

The first step in the pathogenesis of AK is the adhesion of the amoeba to the cornea. This phenomenon is mediated by a series of proteins, the most important being a mannose-binding protein. The process continues with *Acanthamoeba* trophozoites breaking the epithelial barrier by mechanisms of direct cytolysis, phagocytosis, and induction of apoptosis. Subsequently, the amoeba secretes or induces the participation of cysteine, serine, and metalloproteases that degrade the extracellular matrix, basement membrane, and iron-binding proteins and cause a cytopathic effect on host cells. The trophozoites feed on keratocytes and organic particles, causing keratocyte depletion, a marked inflammatory infiltrate, and, finally, stromal necrosis [124,125].

In 2007, Alsam et al. [114] reported the effects of Lf in adhesion assays of *Acanthamoeba* in human corneal epithelial cells. Doses of 0.8–1.48 mg/mL of Lf were incubated with the parasites for 30 min before adhesion assays. The results obtained did not show any statistically significant effect compared to the amoeba without Lf. The authors also used tears from clinically healthy patients and did not observe any difference from *Acanthamoeba* without treatment. It is relevant to know the methods of purification, origin, determination of purity, and amount of iron contained in Lf and in the media used, since its antimicrobial activity depends on the form in which Lf is found (apo or holo) [114]. On the other hand, Lf may not affect adhesion, but it did affect other virulence mechanisms of the amoeba. Only the proteolytic activities of *Acanthamoeba* were tested with tears, but it was not specified whether they were tears from healthy individuals or those with AK—in addition to the lack of testing different doses of purified Lf.

Different results were reported by Tomita et al. in 2017 [115], who examined the viability and encystment of trophozoites of *Acanthamoeba* sp. AA014 from a clinical isolate incubated with different forms of bLf. Here, they use Lf in three forms, apo-bLf, holo-bLf, and native bLf. The iron saturation of these proteins was 2.7%, 97.4%, and 17.9%, respectively. The dose of the three forms of bLf was 10 µM. The tests carried out to determine viability were performed with trypan blue, and a sarkosyl-calcofluor white assay was used for encystment. In all the treatments where apo-bLf was used, they affected parasite viability, caused trophozoite death, and did not allow for the resistance phase of *Acanthamoeba*. Furthermore, apo-bLf did not cause the encystment phase. The other forms of bLf did not show any statistically significant results compared to the control. The results show that the effect may be mainly due to the chelating effect of apo-bLf. Iron plays a key role in the pathogenesis of the disease. However, more trials are needed to determine the amoebicidal mechanism of Lf.

## 5. Conclusions

Parasitic life depends on the ability to obtain nutrients and shelter inside its host. Therapeutic treatments against pathogenic protozoa are often harmful to the host, in addition to the emergence of drug-resistant organisms. Several innocuous compounds from the host itself are objects of study for controlling infectious diseases; among them, natural multifunctional Lf and its peptides derived from the N-terminal sequence present parasiticidal activity in vitro and in vivo. When administered alone, Lf and its derived peptides exhibit potent parasiticidal properties for some pathogens. For others, the effect could be moderate; however, a beneficial effect could be obtained if used in combination with traditional anti-parasitic drugs.

Several mechanisms by which apo-Lf acts against parasites have been described: (1) Lf inhibits parasite growth by sequestering the iron needed for survival; (2) Lf competes for the parasite binding site in the host cell; (3) Lf binds to the parasite cytoplasmic membrane, internal membranes, and cytoskeleton, destabilizing these structures and producing parasite death; (4) Lf exerts its effect in the induction of the immature cyst stage; (5) Lf induces the production of oxygen free radicals; (6) Lf activates macrophages to phagocytize and kill the parasites; and (7) Lf produces parasite death by apoptosis.

Due to the great antimicrobial activity of Lf and functional peptides produced from Lf by the action of proteolytic enzymes—besides any significant adverse effects or intolerance were reported—synergistic effects [46], deriving from the combination of active agents with different modes of action, make Lf an attractive therapeutic option.

More pathways should be further studied to gain a better understanding of the general anti-parasitic mechanism of Lf and its derivative peptides. It is critical to find a specific and common parasiticidal molecule of the active peptide from Lf to all pathogenic protozoan species for the design of future chemicals and the synthesis of anti-adhesion agents to inhibit parasite invasion of the host.

## Figures and Tables

**Figure 1 pharmaceutics-14-01702-f001:**
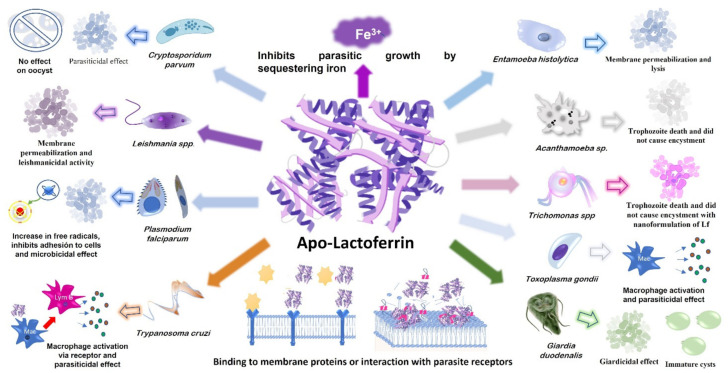
Effect of apo-lactoferrin on pathogenic protozoa. The activity of apo-Lf is to inhibit the growth of infectious agents through iron chelation, pathogen death by membrane destabilization, or parasite neutralization by binding to its receptor. In addition, the activity exerted by apo-Lf is also specified in each pathogen.

**Figure 2 pharmaceutics-14-01702-f002:**
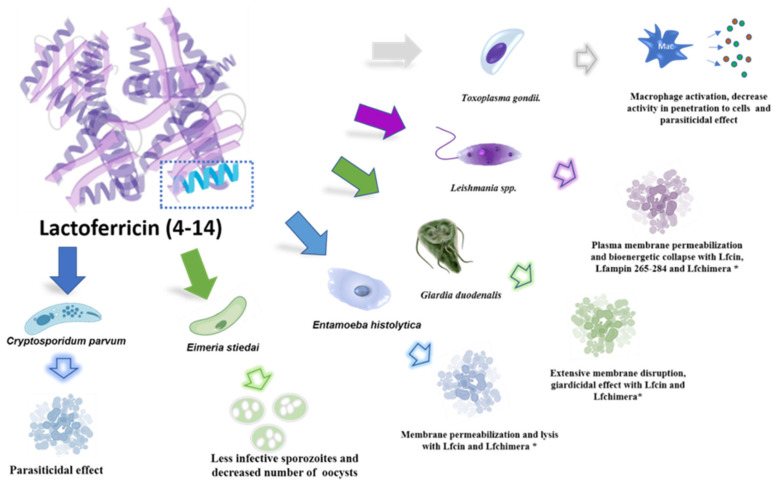
Effect of Lactoferricins on protozoa. Lactoferricins are antimicrobial peptides derived from the N-terminus of Lactoferrin. It has been reported that the effect of Lfcins is higher than that of the native protein. In the image, the activity of Lfcins towards each pathogen is specified. * Lfchimera consists of a fusion of Lfcin 17–30 and Lfampin 265–284.

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
