# Peer review of "Activity of Apo-Lactoferrin on Pathogenic Protozoa"

_pharmaceutics, 2022, doi:10.3390/pharmaceutics14081702_

Round 1
Reviewer 1 Report
Despite this article in the special issue of lactoferrin, the rationale behind the importance of lactoferrin compared to other critical iron binding proteins (e.g. transferrin, ferritin, lipocalin, and various heme-containing proteins) in parasite interactions is not described in the introduction.
Suddenly the authors described digested Lf in 2.1 without reasonable logical development. what is the difference between apo-Lf and N-terminal Lf (apo?) after digestion with pepsin in terms of the efficacy of iron chelation (affinity) and mechanisms (“diverse” in line 68 means nothing to readers).
Lines 69-72 only describes apo-Lf, should be compared to Holo-Lf in these membrane interaction mechanisms. In addition, this paragraph is not well supported by references.
This reviewer anticipated that this entire review primarily focused on the mechanisms of iron metabolism through which Lf interacts and battles with pathogens; however, realized that the rest of the article describes and summarize published data in each parasites and pathogens illustrated in Fig. 1. Unfortunately, these areas are not within this reviewer’s expertise, so no additional comments should be made.
Author Response
Despite this article in the special issue of lactoferrin, the rationale behind the importance of lactoferrin compared to other critical iron binding proteins (e.g. transferrin, ferritin, lipocalin, and various heme-containing proteins) in parasite interactions is not described in the introduction.
We added this paragraph to clarify the purpose of the article
Line 55: This review particularly focuses on the role of apo-lactoferrin and its peptides derivatives in the inhibition of pathogenic protozoa growth, its high affinity for biological membranes, and its lethal effect by disruption of essential membrane functions.
Suddenly the authors described digested Lf in 2.1 without reasonable logical development. what is the difference between apo-Lf and N-terminal Lf (apo?) after digestion with pepsin in terms of the efficacy of iron chelation (affinity) and mechanisms (“diverse” in line 68 means nothing to readers).
The wording was corrected and the peptides derived from Lf were described in detail
Lines 62- 80: In 1992, Bellamy and his coworkers [9] observed that substantial amounts of Lf enter the gastrointestinal tract of mammals as a component of saliva, colostrum, and milk, and ingested Lf appears to have a significant role in the protection of neonates from infectious disease. Posteriorly they have shown that the active peptides generated by pepsin cleavage of human and bovine Lf present bactericidal properties more potent than undigested Lf. Besides, this has led to the identification of the bactericidal domain of Lf near the N-terminus of the Lf molecule. These peptides called Lactoferricins (Lfcins) are well known in the bovine and human forms [10, 11]. Lfcins can pass through the microbial cell membrane and nuclear envelope, suggesting that nucleic acids are a potential target for Lfcins. Lfcins are strongly hydrophobic with positively charged surfaces [12]. Lfcin from bovine is LfcinB17–41, which forms a looped structure through an intramolecular disulfide bond forming a cyclic structure, which is important for its greater antibacterial activity [10, 11]; Human Lfcin [9] is LfcinH1-47, formed by two subfragments connected by disulfide bonds between cysteine residues 1 to 11 and the cyclic residues 12 to 47 [10].
On the other side, based on such common features of antimicrobial peptides, a new peptide was synthesized, the lactoferrampin (Lfampin268–284) from bovine Lf with a broad spectrum of antibacterial activity, which could be isolated using enzymatic activity from bovine Lf. Additionally, human Lfampin was also synthesized (269-285 amino acids) [10, 11]. A fusion peptide between Lfcin 17–30 and Lfampin 265–284 produced and named Lactoferrin-chimera (Lfchimera), formed by 35 amino acids. The binding of these peptides into one molecule resulted in greater antimicrobial activity than each of its peptides and showed a great synergistic activity with different antibiotics [10].
Mechanisms “diverse” in line 68, The wording was corrected in
Lines 87- 92: The mechanisms by which apo-Lf inhibits parasite growth and affects parasite virulence are the theme of a vast literature describing the in vitro or in animal models efficacy of apo-Lf, which are reported below. It has been demonstrated that apo-Lf inter-acts with protozoan membrane cell constituents, such as phospholipids and proteins, destabilizing the membrane and leading to parasite death, as we describe below. The parasiticidal effects of Lf from human, bovine, and buffalo origins have primarily been studied.
Lines 69-72 only describes apo-Lf, should be compared to Holo-Lf in these membrane interaction mechanisms. In addition, this paragraph is not well supported by references.
The activity of holoLf is described in lines (94-107)

Reviewer 2 Report
The present review reports important data for the treatment of pathogenic protozoan, which are usually neglected although their relevance is health.
The manuscript is well written and brings a lot of relevant information, providing fundamental references for scientists studying protozoans and drugs. I highly recommend its publication in Pharmaceutics journal but minor revision should improve its quality and understanding.
1- At the manuscript beggining, a list of pathogenic protozoan could call more attention to the next topics.
2- line 60, provide the meaning of Lfcin
3- Also, it would improve the understanding and make the reading easier introducing a topic or even a table describing the derived peptide forms of lactoferrin and which are their differences
4-Line 89: it is good to mention examples of drugs and protozoans resistant. Also, give examples of natural drugs that have advantages on these organisms (lines 91-92). Cite references.
5- In order to complement Figure 2, a table should improve understanding. It should summarize the protozoan - the disease - the evaluated effects of lactoferrin or derivatives - and the treatment with other drug, if it was tested. 6- Before ending Conclusion, the review should bring a perspective on the therapeutic use of lactoferrin and its derivatives
Those minor revisions will make a great difference in summarizing and making the information easier to visualize in the manuscript, contributing to its use and diffusion among other scientists.
Author Response
The present review reports important data for the treatment of pathogenic protozoan, which are usually neglected although their relevance is health.
The manuscript is well written and brings a lot of relevant information, providing fundamental references for scientists studying protozoans and drugs. I highly recommend its publication in Pharmaceutics journal but minor revision should improve its quality and understanding.
1- At the manuscript beggining, a list of pathogenic protozoan could call more attention to the next topics.
The list of the microorganism included in this work was added to abstract
Lines 11-14: In this work, the activity of Lf against pathogenic and opportunistic parasites such as Cryptosporidium spp., Eimeria spp., Entamoeba histolytica, Giardia lamblia, Leishmania spp., Trypanosoma spp., Plasmodium spp., Babesia spp., Toxoplasma gondii, Trichomonas spp., and opportunistic pathogens Naegleria fowleri and Acanthamoeba castellani were reviewed.
2- line 60, provide the meaning of Lfcin
Line 69: The meaning was added and a description of Lfcins was added
3- Also, it would improve the understanding and make the reading easier introducing a topic or even a table describing the derived peptide forms of lactoferrin and which are their differences
A detailed description was added
Lines 62-80: In 1992, Bellamy and his coworkers [9] observed that substantial amounts of Lf enter the gastrointestinal tract of mammals as a component of saliva, colostrum, and milk, and ingested Lf appears to have a significant role in the protection of neonates from infectious disease. Posteriorly they have shown that the active peptides generated by pepsin cleavage of human and bovine Lf present bactericidal properties more potent than undigested Lf. Besides, this has led to the identification of the bactericidal domain of Lf near the N-terminus of the Lf molecule. These peptides called Lactoferricins (Lfcins) are well known in the bovine and human forms [10, 11]. Lfcins can pass through the microbial cell membrane and nuclear envelope, suggesting that nucleic acids are a potential target for Lfcins. Lfcins are strongly hydrophobic with positively charged surfaces [12]. Lfcin from bovine is LfcinB17–41, which forms a looped struc-ture through an intramolecular disulfide bond forming a cyclic structure, which is im-portant for its greater antibacterial activity [10, 11]; Human Lfcin [9] is LfcinH1-47, formed by two subfragments connected by disulfide bonds between cysteine residues 1 to 11 and the cyclic residues 12 to 47 [10].
On the other side, based on such common features of antimicrobial peptides, a new peptide was synthesized, the lactoferrampin (Lfampin268–284) from bovine Lf with a broad spectrum of antibacterial activity, which could be isolated using enzymatic ac-tivity from bovine Lf. Additionally, human Lfampin was also synthesized (269-285 amino acids) [10, 11].
4-Line 89: it is good to mention examples of drugs and protozoans resistant. Also, give examples of natural drugs that have advantages on these organisms (lines 91-92). Cite references.
Some examples were added
Lines 110- 137: Due to the emergence of pathogenic protozoa resistant to the drugs used as treatments and the side effects that they cause to the patients. Normally, the treatment for pathogens infection is long and with several undesirable side effects, this caused the treatment to be abandoned with the resultant appearance of drug refractory pathogens. In Giardia lamblia, for example, the recent emergence of resistance to the treatment rapidly increased in just a few years (Saghaug, 2019; Leitsch 2017). Metronidazole is a common first-line treatment for giardiasis (Saghaug, 2019; Krakovka), amoebiasis (Upcroft 2001), and trichomoniasis (Cudmore 2004). These conventional drugs however present low efficacy and poor safety. There are no vaccines against major parasitic infections and drugs are the only treatment option. It is necessary to develop other products to combat them.
Natural remedies usually have the advantage of being innocuous, and parasites can show sensitivity without further resistance. These include extracts, fractions, pure compounds, or minerals that are biosynthesized in nature. There are primary and secondary metabolites. Primary metabolites are conserved compounds and necessary for life, while secondary metabolites are not essential for growth but indispensable for survival. Commonly secondary metabolites participate in the defense, protection, and signaling (Monzote 2014). Among several products used, the natural product Artemisin, an antimalarial drug, that was isolated from Artemisia annua, a plant used in traditional medicine (Christen 2001), has been used in the treatment of patients with multidrug-resistant Plasmodium falciparum parasite (Duffey 2021).
Lf is a natural product that has been studied extensively over the past decades. It is a multifunctional protein best known for its ability to bind iron, which eventually led to the discovery of its antimicrobial activity, considered an important host defense molecule.
- In order to complement Figure 2, a table should improve understanding. It should summarize the protozoan - the disease - the evaluated effects of lactoferrin or derivatives - and the treatment with other drug, if it was tested.
A table was included on page 20
- Before ending Conclusion, the review should bring a perspective on the therapeutic use of lactoferrin and its derivatives
The importance of Lf was remarked
Lines 1005 -1008: Due to the great antimicrobial activity of Lf and functional peptides produced from Lf by the action of proteolytic enzymes, besides any significant adverse effects or intolerance, synergistic effects (Leon 06), deriving from the combination of active agents with different modes of action, provide Lf as an attractive therapeutic option.
The English and style were corrected by the American Journal Experts, the certificate was included as pdf archive
